# Steam Refining with Subsequent Alkaline Lignin Extraction as an Alternative Pretreatment Method to Enhance the Enzymatic Digestibility of Corn Stover

**Malte Jörn Krafft [1],[†]** , **Marie Bendler [2],[†]** , **Andreas Schreiber [1]** and **Bodo Saake [1],***

1. Chemical Wood Technology, University of Hamburg, Haidkrugsweg 1, 22885 Barsbüttel, Germany; malte.krafft@uni-hamburg.de (M.J.K.); andreas.schreiber@uni-hamburg.de (A.S.)
2. Thünen Institute of Wood Research, Haidkrugsweg 1, 22885 Barsbüttel, Germany; marie.bendler@thuenen.de
* Correspondence: bodo.saake@uni-hamburg.de; Tel.: +49-40-822-459-206
† Authors of equal contribution.

**Abstract:** Agricultural residues are promising and abundant feedstocks for the production of monomeric carbohydrates, which can be gained after pretreatment and enzymatic hydrolysis. These monomeric carbohydrates can be fermented to platform chemicals, like ethanol or succinic acid. Due to its high availability, corn stover is a feedstock of special interest in such considerations. The natural recalcitrance of lignocellulosic material against degradation necessitates a pretreatment before the enzymatic hydrolysis. In the present study, a novel combination of steam refining and alkaline lignin extraction was tested as a pretreatment process for corn stover. This combination combines the enhancement of the enzymatic hydrolysis and steam refining lignin can be gained for further utilization. Afterward, the obtained yields after enzymatic hydrolysis were compared with those after steam refining without alkaline extraction. After steam refining at temperatures between 160 °C and 210 °C and subsequent enzymatic hydrolysis with Cellic® CTec2, it was possible to enhance the digestibility of corn stover and to achieve 65.4% of the available carbohydrates at the lowest up to 89% at the highest conditions as monomers after enzymatic hydrolysis. Furthermore, the enzymatic degradation could be optimized with a subsequent alkaline lignin extraction, especially at low severities under three. After this combined pretreatment, it was possible to enhance the enzymatic digestibility and to achieve up to 106.4% of the available carbohydrates at the lowest conditions and up to 102.2% at the highest temperature as monomers after following enzymatic hydrolysis, compared to analytical acid hydrolysis. Regarding the utilization of the arising lignin after extraction, the lignin was characterized with regard to the molar mass and carbohydrate impurities. In this context, it was found that higher amounts and higher purities of lignin can be attained after pretreatment at severities higher than four.

**Keywords:** corn stover; pretreatment; steam refining; enzymatic hydrolysis; alkaline extraction; lignin

---

## 1. Introduction

Corn stover is a well-studied agricultural residue of the corn kernel production [1]. It is highly available and a significant amount of the produced straw is undervalued and until now not harvested [2]. Reasons, therefore, are the prevention of soil erosion or the maintaining of soil organic carbon [2,3]. Nonetheless, corn stover is one of the most promising feedstock candidates for large-scale lignocellulose biorefineries [4].

Because of the chemical composition of lignocelluloses, containing high amounts of cellulose and hemicelluloses, the enzymatic hydrolysis (EH) with cellulases is a suitable way to obtain monomeric

carbohydrates for fermentation from the mentioned polysaccharides [3]. The hydrolysis of the glucoside linkage with cellulases (mostly from *T. reesei*) is highly specific to the $\beta$-(1→4) glycosidic bonds of the cellulose chain, resulting in high yields with nearly no undesired by-products formed in the EH. However, inhibitors are normally formed during pretreatment prior to EH.

The chemical composition of corn stover as a lignocellulosic feedstock involves also disadvantages, like the natural recalcitrance of the complex cellulose-hemicellulose-lignin-structure against microbial degradation and EH [5]. One important way to conquer this recalcitrance is the pretreatment of the raw material [6]. Over the years, different methods and applications have been invented and common classifications distinguish between biological, chemical, physical and (hydro)-thermal processes [6].

One additionally described category is the physicochemical pretreatment [7]. Well-described physicochemical pretreatments are steam pretreatments with and without (an acid) catalyst. They are reported as the most often investigated pretreatment methods and they combine the advantages of a simple process, low energy costs and no necessary recycling of process chemicals [7,8].

An alternative physicochemical pretreatment to the generally used steam explosion technique is steam refining [9–14]. Whereas the defibration of the fibers in steam explosion processes is achieved by pressure relief, some studies revealed that the explosion part of the steam explosion is nearly unnecessary for the enhancement of EH yields [15]. Therefore, the defibration can also be achieved by a refining step at the end of the steaming, then called the steam refining process by, e.g., Schütt et al. [12]. Nonetheless, in contrast to the defibration, the dimension of the used biomass particles has shown a much bigger impact on the results of the subsequent EH [12,15]. For steam-refined poplar wood, EH yields increase up to a severity of around four, which represents temperatures around 200 °C and holding times between 10 and 15 min. At severities higher than four the yields decline due to secondary carbohydrate degradation reactions [11]. Unfortunately, degradation of the pentoses to furfural and degradation of the hexoses to 5-hydroxymethylfurfural (5-HMF) occur in processes with increased temperature, like in steam processes. These components have an inhibitory effect on enzymes and it is essential to get knowledge on the best process conditions [16]. However, 5-HMF and furfural are platform chemicals by themselves and can be converted into several value-added products [17,18].

Today, industrial applications of cellulose systems are available at the commercial market for different purposes [19] and the use of nonfood raw material is of high interest. Further, the use of fermentation sugars and the production of platform chemicals from renewable resources are also highlighted in the literature, especially when the applied biorefinery approach is not subject to the food and fuel debate [20–22]. From this point of view, it is favorable to use undervalued agricultural residues, like corn stover, for the renewable production of highly valuable products.

Several authors describe the effect of the lignin content of the used raw material on the performance of the EH. It can be stated that lignin has a negative impact on the yields after EH [23]. Further, the formation of unproductive bindings between lignin and enzymes are described [24]. On the other hand, it was also reported that a nearly complete delignification below 5% lignin content of corn stover with acidified sodium chlorite results in a significant yield loss during the EH, whereas a softer dilute acid pretreatment improves the EH yields [25]. Suggested reasons are the aggregation of the cellulosic microfibrils after the near removal of lignin and xylan and a resulting decreased accessibility. Stücker et al. [13] report further about the utilization of alkaline-extracted poplar lignins in lignin–phenol–formaldehyde resins after a steam refining process. Due to the reported improvement of the enzymatic hydrolysis and the reported possibility of utilization, the influence of an alkaline lignin extraction should be tested for steam-refined corn stover, although alkaline treatment is reported as a preliminary treatment by itself [26].

The aim of the present study was to investigate the effect of different steaming severities on the EH of corn stover. The steamed fibers were subjected to EH with and without alkaline lignin extraction in order to differentiate the effect of lignin on the overall process balance. The degradation products of the carbohydrates were detected to get an overview of inhibitory compounds in the liquid fraction

after the process. Furthermore, characterization of the extracted lignins was performed correlating steaming severities and lignin characteristics.

## 2. Materials and Methods

### 2.1. Raw Material

The used corn stover was harvested in 2018 in Fulda, Hesse (Germany). The material was separated into leaves, stalks, corn cobs and further impurities and was air-dried. After conditioning by air-drying to a stable dry matter content of 90.7%, the material was chopped with a garden chipper into segments with a length between 6 and 8 cm. For steam refining experiments, only leaves and stalks were used.

For raw material analysis, the ash content was measured according to TAPPI standard T 211 om-16. Extractives were determined by Accelerated Solvent Extraction (ASE) of milled (≤1 mm) material with an ASE 350 (Thermo Scientific™ Dionex™, Waltham, MA, USA). Three extraction steps for 10 min at 10 MPa with solvents of different polarities (petrol ether (70 °C); acetone/water 9:1 (70 °C); water (90 °C)) were conducted.

Two-step acid hydrolysis was performed subsequently for the determination of monomeric carbohydrates. Two hundred milligrams of dry material were prehydrolyzed for 60 min with 2 mL of 72% $H_2SO_4$. The reaction was stopped by the addition of 6 mL deionized water and the sample was transferred with 50 mL deionized water into a 100 mL volumetric flask. The second step of hydrolysis was conducted for 40 min at 120 °C and 0.12 MPa overpressure [27].

Afterward, the samples were cooled to room temperature and then filtered through a sintered glass frit (G4). The undiluted filtrate was used for further analysis, described in detail in Section 2.6. The acid-insoluble residue was washed, dried at 105 °C and gravimetrically weighed [12,27].

### 2.2. Steam Refining

The process of steam refining was conducted in a 10 L defibrator (Martin Busch & Sohn GmbH, Schermbeck, Germany). The input of raw material was 200 g dry material. The four blade-refiner-system (illustrated in [28]) in the reactor was rotated only in the final 30 s of steam treatment. The severity factor was calculated with Equation (1) according to Overend and Chornet [29]:

$$\log R_0 = \log \left( t \times e^{\frac{(T-100)}{14.75}} \right) \tag{1}$$

with time in minutes (*t*); temperature in °C (*T*).

For the further understanding of the used process, the different steps of the process are illustrated as a sequential process schematic in Figure 1.

After steaming and refining the raw material, the fiber fraction was washed and the yields were calculated after measuring the solid content. The liquid extract fraction including the wash water was separated and the amount of the combined aqueous extract fraction was gravimetrically measured for further calculations. For calculating the extract yield, the dilution and solid content of the combined extract were gravimetrically measured after freeze-drying. Further process steps will be explained in the following chapters. According to Equation (1), a trial design from 160 °C up to 210 °C was performed with different time steps as outlined in Table 1.

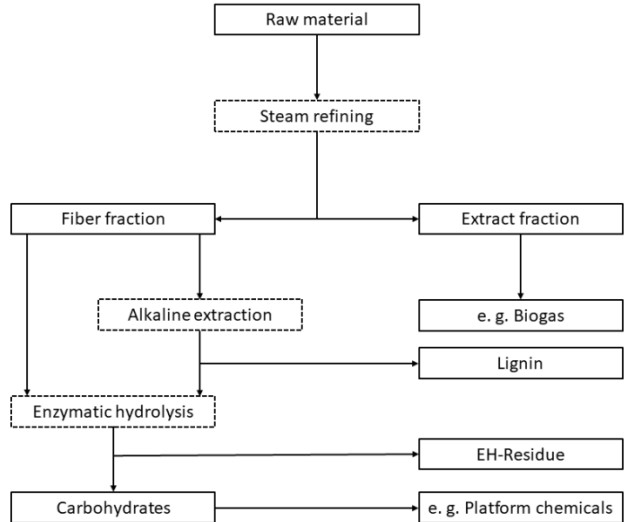

**Figure 1.** Sequential schematic of the used process. Dotted squares representing process steps, solid lines represent products.

**Table 1.** Reaction conditions and corresponding severity factors.

|     | Temperature (°C) | Time (min) | Severity Factor (log $R_0$) |
| --- | --- | --- | --- |
| 1   | 160 | 10 | 2.77 |
| 2   | 170 | 10 | 3.06 |
| 3   | 180 | 10 | 3.36 |
| 4   | 180 | 15 | 3.53 |
| 5   | 190 | 10 | 3.65 |
| 6   | 190 | 15 | 3.83 |
| 7   | 190 | 20 | 3.95 |
| 8   | 200 | 10 | 3.94 |
| 9   | 200 | 15 | 4.12 |
| 10  | 200 | 20 | 4.25 |
| 11  | 210 | 10 | 4.24 |
| 12  | 210 | 15 | 4.41 |
| 13  | 210 | 20 | 4.54 |

### 2.3. Acid Hydrolysis of the Extract and Fibers

The acid hydrolysis of the freeze-dried extracts was conducted as a one-step hydrolysis. One hundred milligrams of dry extractive material were suspended with ultrasound in 10 mL deionized water. Then, 1.8 mL 2N $H_2SO_4$ were added and the suspension was hydrolyzed for 40 min at 120 °C and 0.12 MPa.

For carbohydrate analysis of fibers after steam refining, a two-step hydrolysis (see Section 2.1.) was performed on air-dried and fine milled samples.

### 2.4. Alkaline Lignin Extraction

The lignin extraction was performed according to Klupsch [9] and Schütt et al. [11]. Twenty grams of dry material was treated with 1.6 g (8% of dry raw material) sodium hydroxide and filled up with water to a consistency of 10% solid content. The treatment was carried out at 90 °C for 60 min. The filtered extract was acidified with 6 mL glacial acetic acid to a pH below 4. The precipitated lignin was separated by centrifugation at 18,000× *g* for 10 min.

The fiber residue was washed with hot deionized water and the solid content was determined for yield calculations. The lignin fraction was vacuum dried for 24 h and, afterward, the yields were determined gravimetrically.

## 2.5. Enzymatic Hydrolysis (EH)

The EH of the never-dried fibers was performed with 300 μL cellulases (Cellic® CTec2, Novozymes A/S, Bagsværd, Denmark) and 50 μL β-glucosidases (Novozyme 188, Novozymes A/S, Bagsværd, Denmark) per gram dry material for 72 h at 45 °C and a dry matter content of 4%. The dry matter content was adjusted with a pH 5 phosphate citrate (McIlvaine) buffer. Afterward, the hydrolysate and the fibers were transferred into a 250 mL volumetric flask and were filled up with deionized water. The suspension was filtrated through a sintered glass frit (G4) and the undiluted filtrate was used for carbohydrate analysis.

## 2.6. Analytical Methods

Monomeric carbohydrates in the hydrolysates were determined by borate–anion exchange chromatography (borate–AEC) with a Dionex™ UltiMate™ 3000 (Thermo Fisher Scientific™, Waltham, MA, USA) and MCI GEL® CA08F (Mitsubishi Chemical, Tokio, Japan) as anion exchange resin. Two potassium tetraborate/boric acid-buffers (pH 8.6 and pH 9.5) were used in different concentrations as mobile phase after post-column derivation at 65 °C. Carbohydrates were detected at 560 nm via UV/VIS-spectroscopy. More detailed information about the used borate–AEC was reported by Lorenz et al. [27].

For detection of furfural and 5-hydroxymethylfurfural, reversed phase-high-performance liquid chromatography (RP-HPLC) separation was performed with an AQUASIL™ $C_{18}$ (250 × 4.6 mm; Thermo Fisher Scientific™, Waltham, MA, USA) column for 80 min with 10 μL extract at 25 °C. As a mobile phase, weak acidic water (A; 1 mM $H_3PO_4$) and acetonitrile (B; $C_2H_3N$) were used as eluents in different concentrations and a flow rate of 1 mL/min like shown in Table 2. The detection was conducted at 280 nm.

**Table 2.** Concentrations of the two eluents over time during RP-HPLC.

| Time (min) | c (Eluent A) % | c (Eluent B) % |
|:---:|:---:|:---:|
| 0 | 97.5 | 2.5 |
| 20 | 85 | 15 |
| 50 | 68 | 32 |
| 56 | 62 | 38 |
| 60 | 59 | 41 |
| 63 | 58 | 42 |
| 70 | 58 | 42 |
| 80 | 0 | 100 |

Size exclusion chromatography (SEC) was performed according to Podschun et al. [30] with a mixture of DMSO and 0.1% LiBr as eluent. One guard PolarGel-M column (50 × 7.5 mm; Agilent, Santa Clara, CA, USA) and two PolarGel-M columns (300 × 7.5 mm; Agilent, Santa Clara, CA, USA) were used with a flow rate of 0.5 mL/min$^{-1}$ at 60 °C. Glucose and polyethylene glycol were used as standards with a refractive index detector (RI-501, Shodex™, Munich, Germany). The dissolved samples ($c$ = 1 mg/mL$^{-1}$) were shaken for 24 h at room temperature into the eluent. The sample detection was made with a UV-2077 detector (JASCO, Pfungstadt, Germany) at 280 nm and phenol red as detector matching.

## 3. Results and Discussion

### 3.1. Raw Material Characterization

A comprehensive analysis of the raw material is needed to monitor the effect of steam refining. In the first step, the delivered corn stover was fractionated into its different components and existing impurities. The biggest fractions were leaves (44.4%) and stalks (38.6%). In nearly equal

amounts, impurities (8.8%), mainly sand and corncobs (7.2%), occur. Further, minor amounts of corn silks (0.5%) and corn kernels (0.4%) were contained. In contrast to the determined corn stover composition, other studies report varying results. For example, Pordesimo et al. [31] show for corn stover from Tennessee, USA, a composition of 50.9% stalks, 21% leaves, 15.2% corn cobs and 12.9% husks after excluding the grain fraction. Further, they report good accordance with previous studies. However, corn stover is a natural product and its composition depends on the used variety of maize, the environmental conditions, the harvest time of the raw material and the harvesting technology applied [32].

As described before, the main components of corn stover (leaves, stalks and corncobs) represent around 90% of the delivered material. Not surprising, there is only a small amount of corn kernels left after harvesting the corn with a combine harvester. Nevertheless, a significant amount of sand is included as an impurity. For further investigations, the fractions were separated and the steam refining was performed with the leave and stalk fraction.

Hereafter, the chemical compositions of the used raw material and of pure leaves and stalks were analyzed. The results for the carbohydrate distribution, expressed as monomers, the lignin content, ash and the amounts of extracts, are listed in Table 3.

**Table 3.** Chemical composition of the used raw material in % based on raw material.

|  |  | Raw Material |
| --- | --- | --- |
| Extractives | Petrol ether | 0.8 |
|  | Acetone/Water (9:1) | 8.1 |
|  | Water | 7.3 |
|  | Σ | 16.2 |
| Carbohydrates | Glucose | 35.6 |
|  | Xylose | 19.5 |
|  | Arabinose | 2.9 |
|  | Galactose | 0.9 |
|  | Mannose | 0.3 |
|  | Rhamnose | 0.1 |
|  | Σ | 59.3 |
| Lignin | acid-insoluble [1] | 17.1 [2] |
|  | acid-soluble | 2.2 |
|  | Σ | 19.3 [2] |
| Ash |  | 6.4 |

[1] Mainly analogs to Klason–Lignin [33,34]. [2] Proteins, e.g., from leaves can partly be detected as well in the acid-insoluble residue after hydrolysis.

As mentioned, the main characteristics of the original raw material mix were analyzed by ASE, two-step acidic hydrolysis with the following borate–AEC and determination of the ash content (Table 3). Nonetheless, due to the measuring method, proteins from the leaves may partly be detected in the acid-insoluble hydrolysis residue. Therefore, they are overestimating the detected acid-insoluble lignin content. The insoluble amounts of the raw material might be as well indicating slightly high lignin contents.

However, the found chemical composition is in good accordance with reference values for the different chemical fractions presented in the literature [35–37].

*3.2. Fractions after Steam Refining and Characterization*

3.2.1. Fiber and Extract Yields

Steam refining was conducted in a temperature range from 160 °C to 210 °C with severities from 2.77 to 4.54 (Table 1). To check the statistical scattering, a triplicate at the same severity was tested before. From this previous experiment is known that the standard deviation for the fiber and extract

yields is below ± 1%. After steaming, the solid fiber fraction and the liquid extract fraction were separated, and the yields were calculated after measuring the moisture content of the fiber fraction and the solid content of the extract fraction (Figure 2).

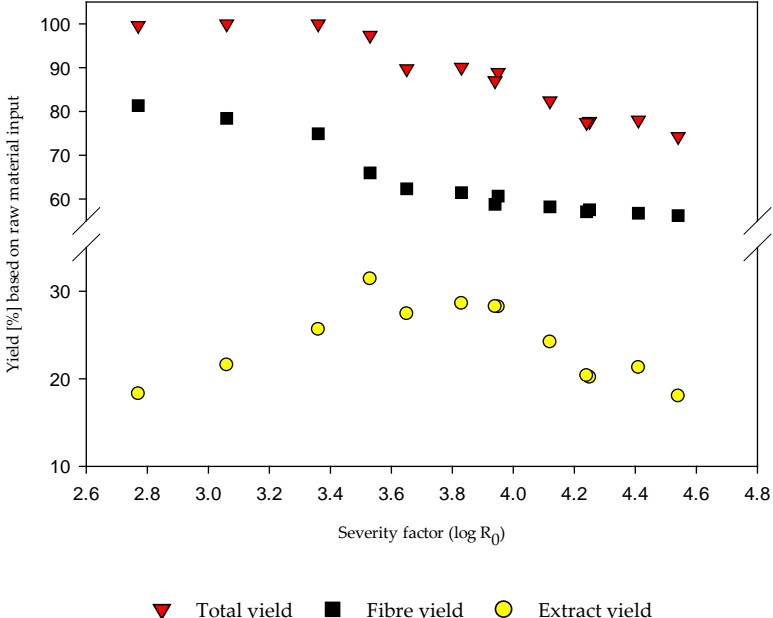

**Figure 2.** Fiber, extract and total yields after steam refining at different severities.

As illustrated in Figure 2 there is a clear tendency for the fiber, extract and total yield with increasing severity of the steam refining. While the fiber fraction yield decreases steadily with enhanced severity, the extract fraction yield shows a maximum at a severity around four. After that point, the extract yield decreases due to degradation reactions of the carbohydrates at severities higher than four. The total yield decreases continuously and at a severity of 4.54, it falls below 75% based on raw material input. The reduced recovery with increasing severity can be attributed to the formation of volatile components, which were not accounted for in the mass balance.

As stated for the fiber yield, extract yields and total yield, there is a decreasing tendency with increasing severity. For mild severity corn stover steam treatments for 10 min at 140 °C (log $R_0$ = 2.18) and 160 °C (log $R_0$ = 2.77), Takada et al. [38] report about slight decreasing yields, as shown in Figure 2. In the present study, only the extracts show a maximum yield at severities around 3.6. Schütt et al. [11] made similar findings for steam refining of poplar wood. In comparison, the fiber, extract and total yields are very similar; however, the described extract maximum occurs at higher severity.

### 3.2.2. Composition of Fibers and Extracts

The carbohydrate and lignin contents in the fiber after steam refining are important process characteristics and figured out in Figure 3. Therefore, the main carbohydrates glucose and xylose, the further hemicellulose monomers (arabinose, galactose, mannose, rhamnose) and as well the acid-insoluble hydrolysis residues were analyzed. The composition was calculated based on the fiber fraction (Figure 3a) and based on the original raw material (Figure 3b).

Regarding the fiber fraction (Figure 3a), it can be stated that with increasing severity the proportion of glucose in the fibers increases. In contrast to that, a strong decreasing tendency is visible for xylose, which is degraded or can be found in the extract fraction due to the preferential solubilization of hemicelluloses. This impact can also be seen for the hydrolysis residue, which is increasing in the fiber fraction due to the loss of xylose and the accumulation of lignin.

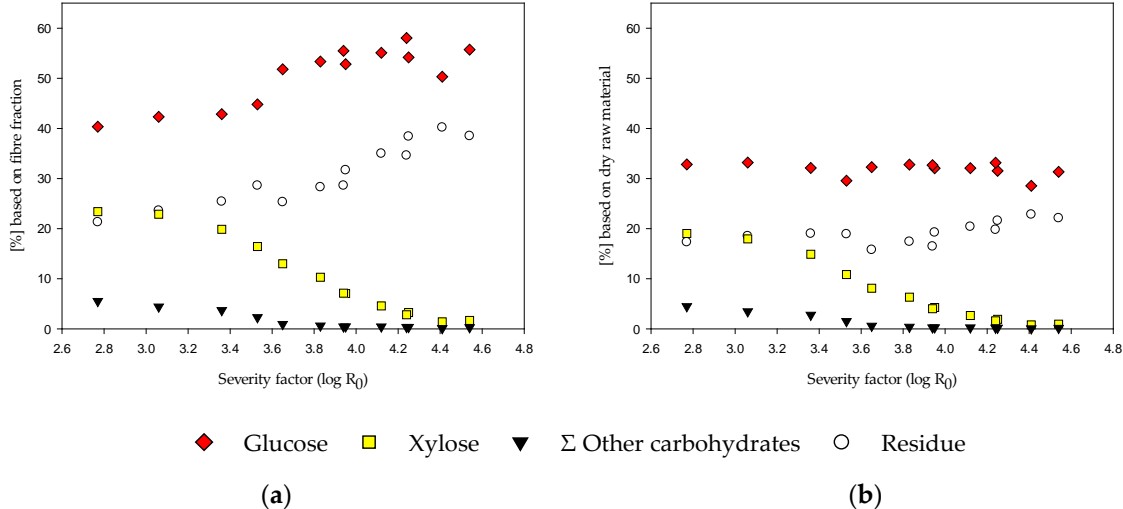

**Figure 3.** Comparison of carbohydrate and residue content of the steam-refined corn stover fiber fraction. The composition of fibers is referred to as fiber fraction (**a**) and raw material input (**b**).

If the obtained fiber yield is considered, the components can be calculated based on the dry material input. With this view, it can be seen that in contrast to the fiber-based view, the amounts of detected glucose and of the hydrolysis residue are nearly stable. Regarding the xylan content, the xylan is heavily degraded or dissolved (Figure 3b) and is thus decreasing.

The described tendencies for the carbohydrates in the fiber fraction, with nearly stable glucose contents and decreasing xylose amounts with increasing severity, are also described in the literature for steam refining of poplar at similar severity factors by Schütt [39]. For the mild steam explosion of corn stover, similar values of lignin, cellulose and hemicellulose are reported recently [38].

Referring to this, Bura et al. [40] described similar findings for steam treatment of corn stover with addition of $SO_2$ at low (log $R_0$ = 3), medium (log $R_0$ = 3.4) and high (log $R_0$ = 4.2) severities. They report increasing glucose yields, decreasing xylose yields and increasing lignin residue for the fiber fraction. Nevertheless, based on raw material the amounts of glucose are in comparison lower, whereas the amounts of xylose in the fiber fraction are much higher for corn stover experiments in contrast to poplar wood results [39].

The extract fraction was analyzed as well regarding the carbohydrate and residue content and is illustrated in Figure 4.

The composition of the extract is referred to as raw material input. When the yield data of the extract are referred to the raw material input (Figure 4), the maximum of xylose obtained between severities between 3.5 and 4 is, of course, less pronounced. This kind of presentation illustrated clearly that rather small quantities of the raw material components can be retrieved in the extract fraction, especially at severities higher than four. However, it is interesting to see that there is a gap between the raw material-based xylose yields from the fibers (Figure 3b) and the extract xylose yields in Figure 4, especially at severities higher than 3.5. This loss of hemicellulose can be explained with the formation of degradation products, like 5-HMF and furfural in the next chapter. Schütt [39] described the extract fraction of steam-refined poplar wood increasing xylose contents up to severities around 4. Subsequently, the xylose degradation occurred due to more intense pretreatment conditions. These findings are quite similar to the findings for the xylose content of the corn stover extracts after steam refining in the present study. Bura et al. [40] report slightly increasing glucose yields and increasing xylose yields in the extract fraction after steam pretreatment of corn stover at low (log $R_0$ = 3), medium (log $R_0$ = 3.4) and high (log $R_0$ = 4.2) pretreatment conditions. They did also report a slight increase of the xylose in the extract fraction from a severity of 3.4 to 4.2. It must be remarked that the authors used $SO_2$ as acid catalysts for the dataset. Additionally, the medium pretreatment condition was used without an acid catalyst. Higher yields of xylose in the extract fraction at medium conditions

are clearly visible for the experiments with $SO_2$, but by calculating the total xylose mass balance there is a higher xylose loss visible compared with the results without a catalyst. However, the results without an acid catalyst are comparable with the results presented in this study.

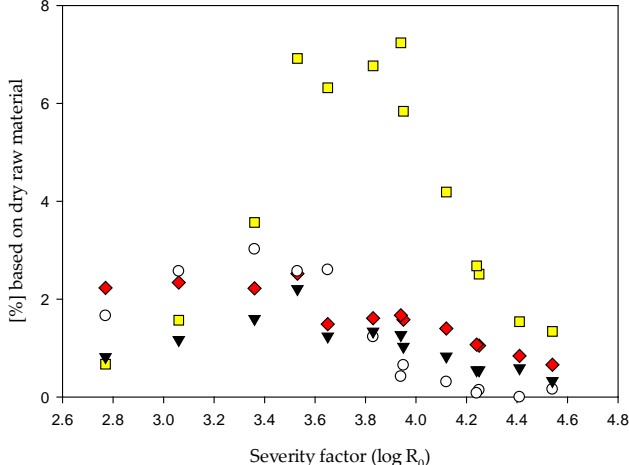

**Figure 4.** Comparison of carbohydrate and residue content of the steam-refined corn stover extract fraction.

### 3.2.3. Detection of 5-HMF, Furfural and pH Value

The negative influence of 5-HMF and furfural as main carbohydrate degradation products on enzymatic hydrolysis and subsequent fermentation is well described [41–44]. Therefore, 5-HMF and furfural were analyzed. As these compounds are unstable, the detection was performed directly in the extract obtained after the steam refining of the corn stover. The results are shown in Figure 5.

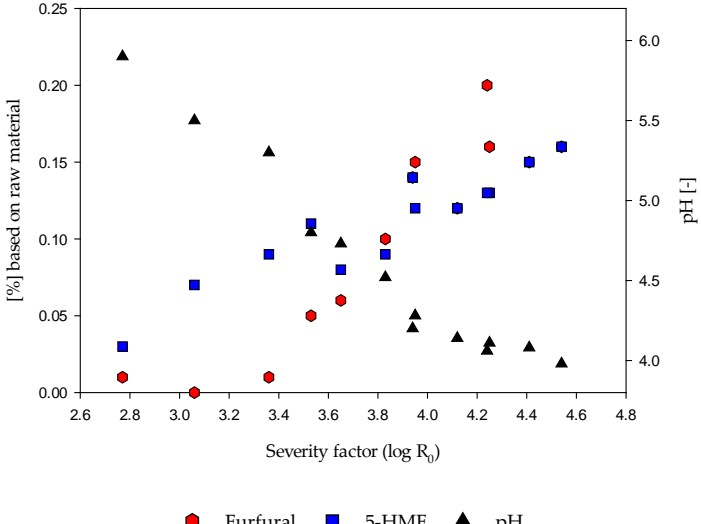

**Figure 5.** Effect of steaming at different severities on pH and the amounts of 5-HMF and furfural in the extract fraction.

Figure 5 shows that both, the content of furfural and 5-HMF in the extract fraction, increased steadily with increasing severity factors. This has particular importance for the approach to saccharify the fibers and the extract simultaneously. However, it is also important for subsequent processes such as EH of the fibers to fermentable carbohydrates or the production of biogas from the extract. The degradation of hexoses and pentoses to 5-HMF and furfural and the lowering of the pH value during steam explosion and steam refining is also well-described in the literature [11,45–49].

Ruiz et al. [49] report almost similar results for 5-HMF and furfural after steam explosion of sunflower stalks in comparison with the presented results for corn stover. Therefore, it can be assumed that the lower furan contents are typical for agricultural residues. Jacquet et al. [47] report results for really high severities up to 5.56 at extreme temperatures around 260 °C. For severities below severities of 4.5, they report similar data to the present data, although they used microcrystalline cellulose. For further results, a strong increase of 5-HMF, but not for furfural, is reported. In this context Um and van Walsum [50] report about the formation of furfural with increasing severity by a simultaneous decrease of the xylose contents in a corn stover dilute acid pretreatment. However, for severities above 4.43, they report a decrease and degradation of furfural contents and an increase of formate. In contrast to the present data, they received high severities by time, not by temperature.

However, the measured values for 5-HMF and furfural after steam refining of corn stover are much lower than the measured values for the furans after steam refining of poplar wood [11].

Figure 5 depicts as well the pH value in the extracts, which decreases with increasing severity. This is due to the formation of formic and acetic acid. Formic acid concentrations ranging from 0.61% at a severity factor of 3.65% up to 1.53% at a severity factor of 4.54%, all values based on raw material. The corresponding data for acetic are 2.02% and 2.89%, based on raw material.

Due to the formation of these organic acids, mainly caused by hemicellulose degradation, the pH value is reduced and autohydrolysis is intensified.

### 3.3. Alkaline Lignin Extraction

#### 3.3.1. Fiber and Lignin Yields

The extraction of lignin with 8% NaOH (based on raw material input) was conducted for the fiber fraction of all severity grades. The loading of 8% NaOH is reported as the optimum for alkaline extractions after steam refining [9,11] and represents at a consistency of 10% a thin 0.8% *w/w* NaOH solution. The total recovery rate was calculated with the lignin and fiber recovery after extraction and is illustrated in Figure 6.

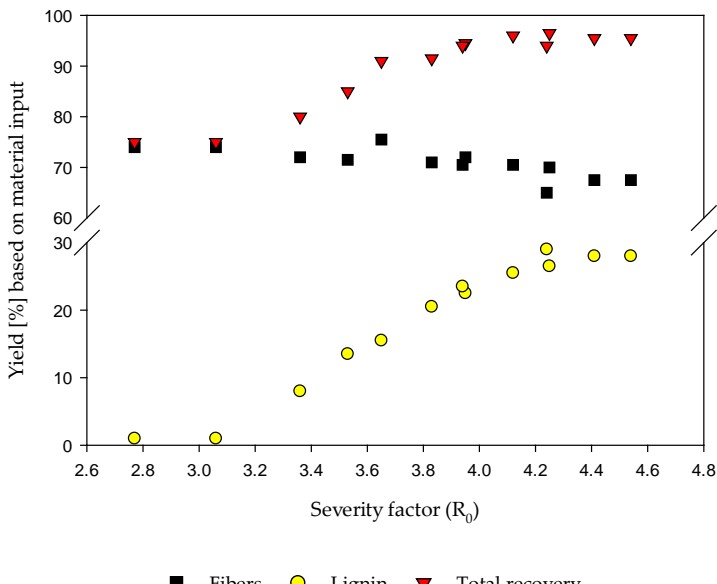

**Figure 6.** Illustration of the fiber and lignin yield and the total recovery rate, containing both mentioned fractions, after alkaline lignin extraction.

The fiber yield is slightly, but continuously decreasing with increasing severity (Figure 6). Simultaneously the lignin yield increases with the severity. Surprisingly, the reduction of fiber yield is significantly lower compared to the increase of lignin yield. This might be due to the condensation

reactions of the lignin at high temperatures. The lignin with higher molar masses can be precipitated easier or more efficiently resulting in higher yields. However, when there is no covalent binding to the lignin, the precipitation is not sufficiently possible, resulting in lower yields.

As further illustrated, the combined recovery of lignin and fibers after extraction is strongly increasing with increasing severity of the pretreatment (Figure 6) from 75% up to 95% of the original raw material input. The increase in the total recovery rate can mainly be attributed to the higher lignin precipitation at severities above 4.

### 3.3.2. Analysis of the Lignins

The effect of steaming severity on the molar mass of the precipitated lignins is illustrated in Figure 7. It becomes evident that the molar mass of the lignins increased gradually with intensifying the steam refining conditions. Furthermore, the main peak of the molar mass distribution is visible between 6800 and 7700 g/mol (Figure 7).

At a severity of 3.65 (190 °C/10 min), components with low molar masses (100–250 g/mol) are visible in the distribution curves. They are significantly reduced at severities above four. The increasing shoulder at high molar masses is a clear indication for condensation reactions at higher severities. Thus, the SEC results confirm the assumption that lignin condensation occurs to a higher extent under harsh steaming conditions.

For considerations on the utilization potential of the extracted lignins, knowledge of the lignin characteristics and purity is essential. Therefore, their carbohydrate contents were determined after acid hydrolysis (Figure 8). With increasing severity, the carbohydrate impurities of lignin decrease. Nevertheless, after pretreatment at lower severities than 3.95, more carbohydrate impurities can occur in the alkaline extract. At severities above four, the carbohydrate content of lignins is negligible. For this finding, Schütt et al. [11] and Schütt [39] also report decreasing carbohydrate impurities with increasing severity for extracted lignins of steam-refined poplar wood.

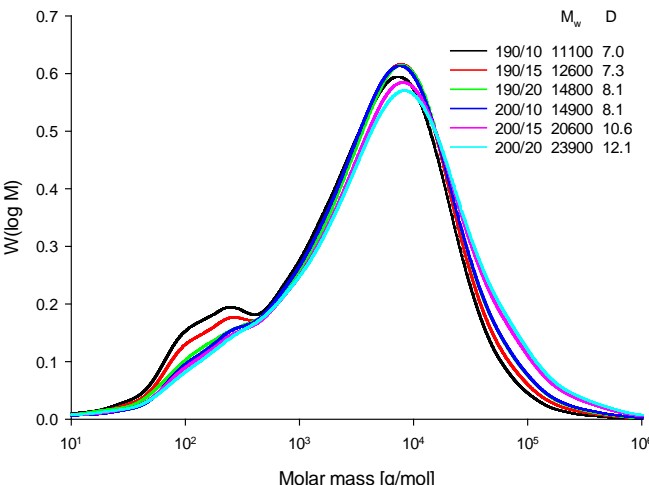

**Figure 7.** Molar mass distribution, molecular weight ($M_W$) and dispersity (D) of alkaline-extracted lignin.

Several authors discussed the alkaline extraction of steam-treated fibers and following the influence of lignin removal on EH. For the process of lignin extraction, Schütt et al. [11] and Schütt [39] reported higher molar masses and a higher dispersity of the extracted lignin with increasing severity. The influence of one or two steam explosion steps on the extraction behavior of the lignin and the influence of $SO_2$ as a catalyst were also investigated by Li et al. [51]. However, the reported findings are in good accordance with the illustrated results in the present study (Figures 7 and 8).

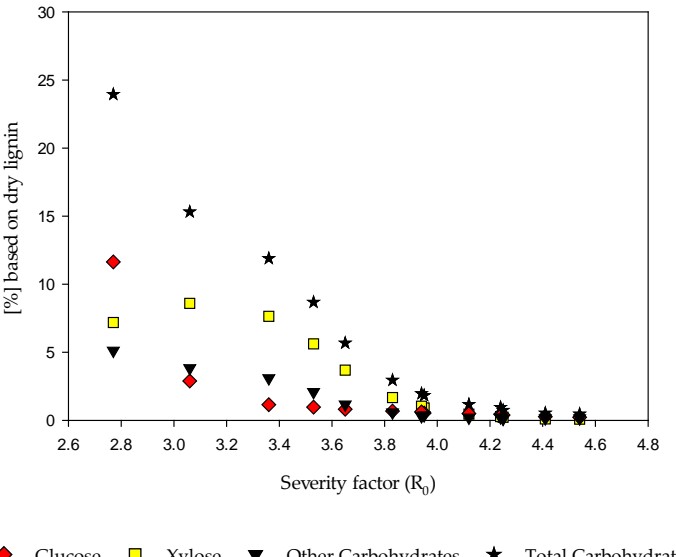

**Figure 8.** Effect of steaming severity on the carbohydrate content of alkaline-extracted lignins.

### 3.4. EH of the Fiber Fraction with and without Alkaline Extraction

The enzymatic hydrolysis of steam-refined fibers was compared with and without alkaline extraction. To enable a detailed comparison all carbohydrate yields were calculated on the theoretically available carbohydrates in the used fibers, detected after two-stage acidic hydrolysis (Figure 9). Therefore, the detected monomeric carbohydrates were concerned as a percentage based on the carbohydrate content in the fibers after an EH with Cellic® CTec2.

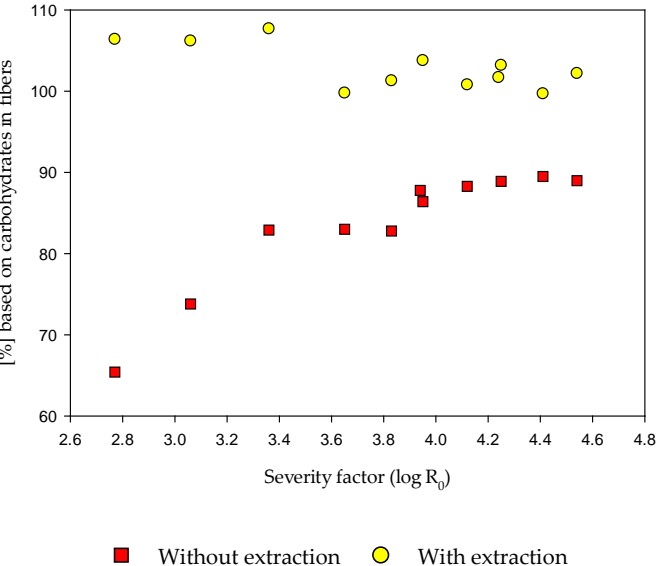

**Figure 9.** Effect of severity and alkaline extraction on the enzymatic hydrolysis (EH) of the fibers with Cellic® CTec2. The EH yields are calculated based on the theoretically available carbohydrates in the used fibers.

It is obvious that without extraction of the lignin, the yields after EH increases strongly with increasing steaming severity. After alkaline lignin extraction, there is no more positive effect of severity. There are slightly higher yields at severities lower than 3.5 and yields surpass constant 100%. This is because available carbohydrates in the fiber fraction were determined with a two-step acid hydrolysis. In this process, the carbohydrate content can be underestimated due to the secondary degradation of

carbohydrates in the analytical method. This degradation will be more pronounced at severities below around 4 because more hemicelluloses and amorphous cellulose are present in the fiber. Hydrolysis with enzymes is more selective to carbohydrates and results in fewer degradation products. Therefore, higher yields are explainable by the high efficiency of EH. It can be concluded that after steam refining and alkaline extraction, all carbohydrates are available for the EH.

The described improvement of the EH yields by steam refining is an undisputed fact and pretreatments are often regarded as indispensable prerequisites for the economic implementation of such processes [6]. There are several parameters influencing the EH. Important characteristics for the impact on the EH are, e.g., the ratio of amorphous and crystalline regions, the degree of polymerization (DP) of the sample, the moisture content, available surface areas, the lignin content or the pore size of the samples [52]. Higher yields after EH are reported for wet substrate in contrast to dried biomass [53], as also found in this study (data not shown here). However, the effect of lignin removal is controversially discussed in the literature. Some authors complained about studies on wood having a reduced enzymatic digestibility after lignin extraction due to collapsing pores [54–56]. Schwalb et al. [57] reported no influence of the alkali extraction, whereas Excoffier et al. [58] and Schütt et al. [11] presented higher yields of the EH after lignin removal. Ishizawa et al. [25] report that a nearly complete delignification with acidified sodium chlorite below 5% lignin content decreases the yields after EH strongly. The authors report further that the influence of partial lignin and xylan extraction improves the EH. However, the xylan removal is suggested to be more significant for the improvement of the EH yields than lignin removal [25].

Regarding the discussions in the literature, a positive influence of the partial lignin removal on the EH results could also be stated in the present study for corn stover. Furthermore, the higher impact at severities below 4 and the fading influence at severities above 4 (Figure 10a,b) could be confirmed in accordance with Schütt et al. [11]. For more findings of the overall process efficiency, the glucose (Figure 10a) and xylose yield (Figure 10b) based on raw material were calculated for the process variants with and without lignin extraction prior to EH. Taking the yields after steam refining into account a strong influence of the extraction is apparent at severities below 4. This effect is disappearing at severities >4. After that point, no significant difference can be determined for the two process variants. Similar findings were reported for the steam refining of poplar by Schütt et al. [11]. The described influence at severities lower than 4 is also visible for the xylan yields. However, the influence is not that strong as for the glucose yields.

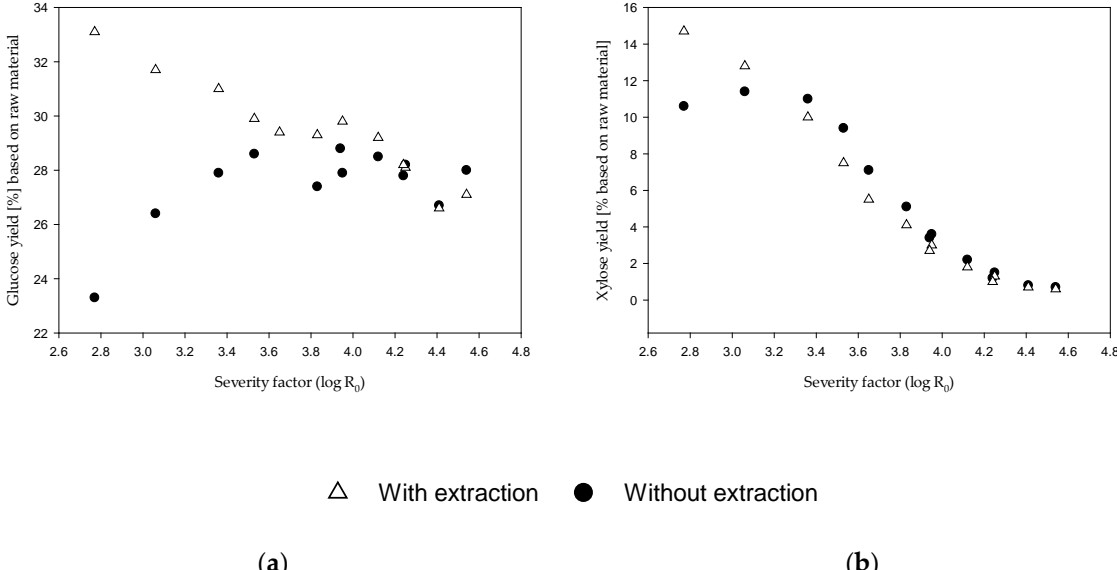

**Figure 10.** Glucose yields (**a**) and xylose yields (**b**) after EH based on raw material for the process variants with and without alkaline extraction prior to EH.

*3.5. Overall Process Balances*

Finally, the glucose and xylose yields after EH of the fiber fraction and glucose and xylose yields after one-step acidic hydrolysis of the extract fraction were compared and calculated based on raw material. This consideration was made in order to evaluate the effect of alkaline extraction of the overall process balance and to monitor the carbohydrate losses during the whole process (Figure 11). Schütt [39] reports a consideration that a severity of around 4.5 is needed for gaining sufficient glucose rates after steam refining of poplar wood and enzymatic hydrolysis with Celluclast$^\circledR$.

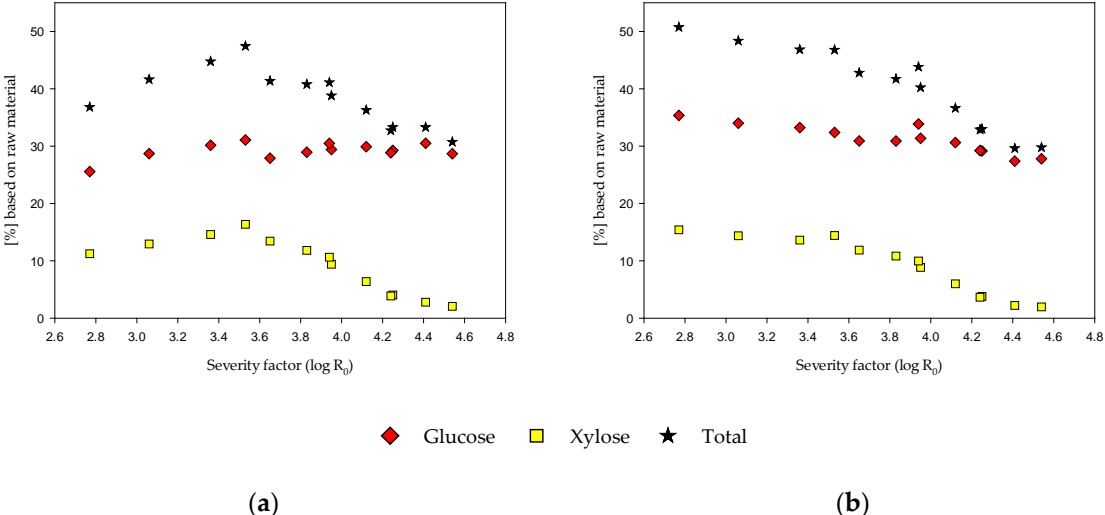

(**a**)              (**b**)

**Figure 11.** Summed glucose and xylose yields after EH of the fiber fraction and after one-step acidic hydrolysis of the extracts—without (**a**) and with (**b**) alkaline extraction; all based on raw material input.

A different behavior was found for the steam refining of corn stover and enzymatic hydrolysis with Cellic$^\circledR$ CTec2 in this study. It must be remarked that the following comparisons and differences are mainly due to the used enzymes. From previous experiments (data not shown here) it is clearly known that Cellic$^\circledR$ CTec2 shows a much higher activity to the used biomass than Celluclast$^\circledR$, used in previous days. However, not shown comparison experiments with Celluclast$^\circledR$ and steam-treated corn stover also show the highest glucose yields at severities around 4.5.

As shown in Figure 11a the glucose is also available at severities below 3.5 and there is no need for steam refining in higher regions. There is a slight increase in the glucose yield with increasing severity with no significant optimum. In contrast to those findings, there is a clear optimum for the xylose yields at a severity of 3.2. Regarding Schütt [39], the findings for the xylose yields are nearly equal and the optimum is also in severity regions below 3.5. Figure 11b represents the overall process balance after alkaline extraction of the fibers and following EH. In contrast to the prior findings, the optimum of all yields is now located at a severity of 2.77. Due to these findings, the optimal steaming conditions for steam-refined and steam-extracted samples are clearly visible at severities below 3.5, mainly due to the higher reactivity of the used enzymes.

## 4. Conclusions

Due to the findings in the present study, several conclusions can be made. For steam refining experiments without subsequent alkaline extraction, the optimum of EH yields is located at a severity around 3.95. However, the optimum of total carbohydrate recovery from EH of the fibers and acidic hydrolysis of the extract fraction is at a severity around 3.4 with around 47.5% based on raw material.

For steam refining with subsequent alkaline extraction, different findings were made. For the EH yields around 100% of the theoretically available carbohydrates were found, even at severities below 3.5. However, also the total carbohydrate recovery shows the highest yields at these severities

and it is known that low contents of carbohydrate degradation products are beneficial for further process steps. Nonetheless, for lignin utilization severities around 3.95 might be optimal due to fewer carbohydrate contaminations.

Due to these findings, it can be concluded that steam refining pretreatment especially at severities below 3.5 seems to be interesting for corn stover. It could further be stated that steam refining at different severities and alkaline lignin removal is enhancing the enzymatic digestibility for the supply of monomeric carbohydrates significantly. Higher fiber yields and good digestibility after alkaline extraction giving high yields of fermentable carbohydrates for further value-added products, like ethanol or dicarboxylic acids. Therefore, further undervalued agricultural residues should be tested in the future at severities below 3.5 and with alkaline extraction. Afterward, the results can be compared with the results for corn stover presented in this study.

**Author Contributions:** Conceptualization, B.S.; funding acquisition, B.S.; investigation, M.J.K., M.B. and A.S.; supervision, B.S.; visualization, M.J.K.; writing—original draft, M.J.K.; writing—review and editing, M.J.K., M.B., A.S. and B.S. All authors have read and agreed to the published version of the manuscript.

**Funding:** This research was funded by the Federal Ministry of Education and Research (BMBF) via Project Management Jülich (PtJ) in the PANDA project, grant number 031B0505B.

**Acknowledgments:** The Federal Ministry of Education and Research (BMBF) and Project Management Jülich (PtJ) are gratefully acknowledged for their financial support. The authors wish to thank Othar Kordsachia, who assisted in the proofreading of the manuscript. Martin Bellof, Autodisplay Biotech GmbH, is thanked for the raw material acquisition. Anna Knöpfle, Nicole Erasmy and Sascha Lebioda are also gratefully acknowledged for their technical support.

**Conflicts of Interest:** The authors declare no conflicts of interest.

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
