# Peer review of "Steam Refining with Subsequent Alkaline Lignin Extraction as an Alternative Pretreatment Method to Enhance the Enzymatic Digestibility of Corn Stover"

_agronomy, doi:10.3390/agronomy10060811_

Round 1

Reviewer 1 Report

This manuscript investigated the enzymatic hydrolysis of a combined steam refining process and alkaline lignin extraction process of corn stover.  Results showed that this combined process could enhance the enzymatic digestibility of corn stover. The experimental design is good and the results are fine. 

  1. Abstract & Keywords: 
    1. Please include some key numerical results, such as the % of enzymatic digestibility of combined pretreatment and steam refining only process in the abstract section. 
    2. CTec2 is not needed in the keywords section.
  2. Introduction:
    1. Line 58-59:'for steam-refined poplar wood, EH yields increase up to a severity of around 4.' Given readers may not be familiar with the degree of severity (4 could be either high, low, or moderate), please provide more details on this sentence.  
    2. Line 71-80: authors did a lit review on acid pretreatment on lignocellulosic biomass and then 'due to these findings, the influence of an alkaline lignin extraction should be tested,' why? Since alkaline pretreatment itself could be used to pretreat corn stover, then why combining it with the steam refining process? The knowledge gap is not fully addressed by the authors. 
  3. Materials and Methods:
    1. What are the criteria for selecting temperature and time in the experimental design? 
  4. Results and Discussion
    1. In 3.1, there is no need to show the composition results of pure leaves and stalks since corn stover is the raw material in this study, not leaves of corn stover or stalks of corn stover. 
    2. In Fig.1, please keep the major unit of y-axis constant below and above the axis break. 
    3. In 3.2.1, is there any statistical analysis conducted on these yields? are they significantly different from the statistical view?
    4. Fig.2 is confusing. Suggest focusing on fiber part only. No need to illustrate extracts in the main text (could move extract-related analysis to SI if necessary).
    5. Line 281-282, well described by who? please add references. 
    6. Suggest using error bars in all figures to show the variation of the results. 
    7. In Fig.7, from what I eyeballed, nearly all EH yield with alkaline extraction surpassed 100% and below 110% under various severity factors. Is the EH yield of lower severity factors significantly different from that of a higher severity factor? 
    8. What do the shaded areas mean in Fig.7 and Fig. 9?
    9. What is the impact of sodium hydroxide loading per dry g of biomass on EH yield?
    10. In the economic analysis of cellulosic biorefineries, the pretreatment is reported to be one of the most capital-intensive processes; in this study, the authors used a combined steam refining process and an alkaline process, although the EH yield increases, will this combined process be economically available?  

Reviewer 2 Report

Dear Authors,

The topic for your study is novel and relevant for biomass refining. 

However, there is room for improvement in the quality of the manuscript. 

I have left most of my comments in the attached manuscript. The major points of improvement are:

  1. I am missing the description of steam refining.
  2. How is steam refining different from steam explosion.
  3. A sequential process schematic of the steps involved in the experimental study would significantly improve the understating of your results.
  4. The equation for severity is incorrect.
  5. Throughout the manuscript the term low severity and high severity is used. These are ambiguous terms. You must describe quantitatively what you consider as high, moderate, and low severity.
  6. I am missing the dry matter content in your results and analysis.
  7. Figure 2 and its supporting description can be improved.

Overall I believe, that you have explored an interesting topic of steam refining coupled with alkaline lignin extraction to enhance carbohydrate yield from lignocellulosic biomass.  

Reviewer 3 Report

Although the authors presented plenty of data, the novelty of this research is not coming across clearly. I suggest a clear statement of novelty in the introduction before the objectives.

Provide the details of air drying before chopping.

Can the authors provide more discussion on the effect of severity on detection of inhibitors (5-HMF, furfural, and others)?

Round 2

Reviewer 1 Report

The manuscript has been revised according to the reviewers' suggestions.

Reviewer 3 Report

The manuscript is better now.